# Latent space Projection Predictive Inference

## Abstract

Given a reference model that includes all the available variables, projection predictive inference replaces its posterior with a constrained projection including only a subset of all variables. We extend projection predictive inference to enable computationally efficient variable and structure selection in models outside the exponential family. By adopting a latent space projection predictive perspective we are able to: 1) propose a unified and general framework to do variable selection in complex models while fully honouring the original model structure, 2) properly identify relevant structure and retain posterior uncertainties from the original model, and 3) provide an improved approach also for non-Gaussian models in the exponential family. We demonstrate the superior performance of our approach by thoroughly testing and comparing it against popular variable selection approaches in a wide range of settings, including realistic data sets. Our results show that our approach successfully recovers relevant terms and model structure in complex models, selecting less variables than competing approaches for realistic datasets.

## 1 Introduction

Variable and structure selection plays an important role in a robust Bayesian workflow (Gelman et al., 2020). While variable selection has been extensively studied and successfully applied for models in the exponential family (Koopman, 1936), it has not received the same attention regarding models outside the exponential family (e.g., advanced ordinal, count or time-to-event (also known as survival) data distributions), despite their important applications in different fields (Kelter, 2020; Nagler, 1994; Bürkner & Vuorre, 2019; Barron, 1992).

We propose an efficient, stable, and information theoretically justified method to make variable selection for non-normal observation models in or beyond the exponential family, while retaining the predictive performance of the full model. The main benefits of the proposed *latent space projection predictive inference* are:

1. We enable the projection predictive variable and structure selection for models outside the exponential family, while honouring the original model structure and its predictive uncertainty.
2. We obtain more stable projections for non-Gaussian exponential family models.
3. We demonstrate the superior performance of our method as compared to state-of-the-art competitors in both simulated and real-world scenarios.
4. We provide a ready-to-use open source implementation of the new methods.

## 2 Projection predictive inference

We assume a modeller has built a rich model $p(y|X, \lambda)$ with prior $p(\lambda)$ to predict outcome $y$ given predictor variables $X$ and parameters $\lambda$. The model has passed model checking, and now the modeller wants to find a simpler submodel with a similar predictive performance. The rich model is called a reference model. Given posterior draws $\{\lambda_*^{(s)}\}_{s=1}^S \sim p(\lambda \mid \mathcal{D})$ from the reference model with data $\mathcal{D} = \{X, y\}$, projection predictive inference (Piironen et al., 2020b; Catalina et al., 2022; McLatchie et al., 2023) learns a projection $q_\perp(\lambda)$ containing only a subset of variables that matches the reference predictive performance as close as possible. The solution is given by the minimiser of the Kullback-Leibler (KL) divergence from the reference model to

the projection predictive distributions (Dupuis & Robert, 2003). Let $p(\tilde{y} \mid \mathcal{D})$ be the reference predictive distribution, $\lambda_\perp$ be the parameters of a submodel (subset of predictor coefficients), and $\{\lambda_\perp^{(s)}\}_{s=1}^S \sim q_\perp(\lambda)$ be draws from the projection, and $q_\perp(\tilde{y}) = \int p(\tilde{y}|\lambda_\perp) q(\lambda_\perp) d\lambda_\perp$ be the projection predictive distribution, then:

$$\mathrm{KL}\left(p\left(\tilde{y} \mid \mathcal{D}\right) \| q_\perp\left(\tilde{y}\right)\right) = -\mathbb{E}_{\lambda_*}\left(\mathbb{E}_{\tilde{y}|\lambda_*}\left(\log \mathbb{E}_{\lambda_\perp}\left(p\left(\tilde{y} \mid X, \lambda_\perp\right)\right)\right)\right) + \mathrm{C}. \tag{1}$$

As the integrals involved are in most cases computationally infeasible, Goutis & Robert (1998) suggested approximating Eq. (1) by changing the order of integration and minimisation and solve the optimisation for each posterior draw separately

$$\arg\max_\lambda \mathbb{E}_{\tilde{y}|\lambda_*^{(s)}}\left(\log \mathbb{E}_{\lambda_\perp}\left(p\left(\tilde{y} \mid X, \lambda\right)\right)\right). \tag{2}$$

With this approach, $q(\lambda_\perp)$ and $q_\lambda(\tilde{y})$ are never constructed explicitly, but approximated using the projected draws $\{\lambda_\perp^{(s)}\}_{s=1}^S$.

For models in the exponential family distribution, Eq. (2) coincides with computing maximum likelihood estimates under the projection model as

$$\lambda_\perp = \arg\max_\lambda \sum_{i=1}^N \mu_i^* \xi_i(\lambda) - B(\xi_i(\lambda)), \tag{3}$$

where $\mu^* = \mathbb{E}_{\tilde{y}|\lambda_*}(\tilde{y})$ are mean predictions of the reference model, $\xi_i$ are the natural parameters for the $i$th observation and $B(\cdot)$ is a function of the natural parameters (McCulloch, 2003). These maximum likelihood estimates can be efficiently computed by solving penalised iteratively reweighted least squares (PIRLS; Marx, 1996). If non-constant, the projected scale parameter of the exponential family model is then obtained as

$$\phi_\perp = \arg\max_\phi \sum_{i=1}^N \left(\frac{r_i(\lambda_\perp)}{A(\phi)} + \mathrm{E}_{\tilde{y}_i|\lambda_*}(H(\tilde{y}_i, \phi))\right), \tag{4}$$

where $A, H$ are family-specific functions and $r_i(\lambda_\perp) = \mu_i^* \xi_i(\lambda_\perp) - B(\xi_i(\lambda_\perp))$ does not depend on $\phi$.

Piironen et al. (2020b) proposed a further speed-up by first clustering the posterior draws $\lambda_*^{(s)}$ and then solving the optimisation individually for each resulting cluster centre $\{\lambda_*^{(c)}\}_{c=1}^C$.

Summary of the variable selection using projection predictive approach: 1) fit the reference model with MCMC and store posterior draws $\{\lambda_*^{(s)}\}_{s=1}^S$, 2) for each submodel that is considered and each posterior draw find the projected draw $\lambda_\perp^s$, 3) form the predictive distributions based on $\{\lambda_\perp^{(s)}\}_{s=1}^S$. Given the projected submodel predictive distributions, 4) search through the model space (e.g. with forward search) to find the best projected model for each model size, and 5) use cross-validation to use to choose the smallest model size having similar predictive performance as the reference model (see more details in Piironen et al., 2020b; McLatchie et al., 2023). The use of reference model and projection significantly reduce the model selection variance and selection induced overfitting (Pavone et al., 2022). In case of exponential family models, projection predictive approach has been shown to outperform 10-fold cross-validation, WAIC, DIC, maximum a posteriori (MAP), median marginal posterior probability model, $\mathrm{L}^2/\mathrm{L}_{CV}^2/\mathrm{L}^2-k$-criteria, Lasso, Lasso-relaxed, Elastic net, and Ridge regression (Piironen & Vehtari, 2017a; Piironen et al., 2020b).

## 3   Latent space projective inference

The equivalence between maximum likelihood estimates and KL minimiser does not hold for models outside the exponential family. For data $\mathcal{D} = \{X, y\}$, parameters $\lambda$ and inverse link function $g$, we assume the general model formulation

$$y \sim p\left(\mu, \phi\right), \quad \mu = g(\eta), \quad \eta \sim p\left(\eta \mid \lambda, X\right), \tag{5}$$

where $\mu$ is a location parameter depending on predictors $X$, $\phi$ is a global dispersion (shape) parameter, $g$ is a link function, and $\eta$ is the latent predictor which is often linear ($\eta = X\lambda$; McCullagh & Nelder, 1989).

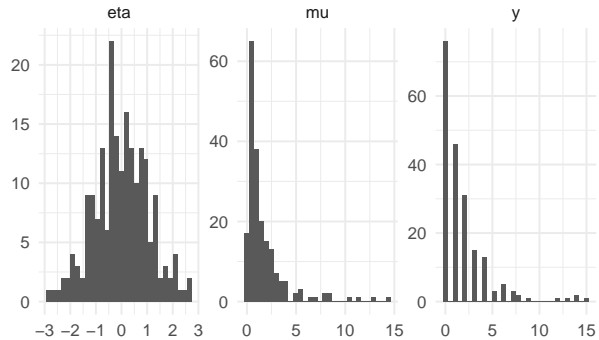 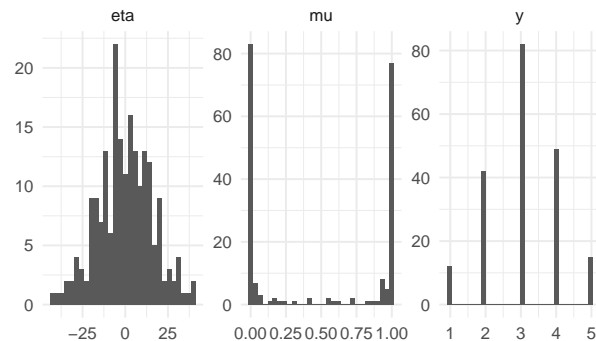

(a) Latent linear predictor, transformed predictor and response space for a Poisson model with `log` link function.

(b) Latent linear predictor, transformed predictor and response space for an ordinal cumulative model with `probit` link function.

Figure 1: Latent ($\eta$), transformed ($\mu$), and response ($y$) space representation for two models.

Fig. 1 shows the difference between the linear and the transformed predictor spaces for two common models, a Poisson model with a `log` link function (no $\phi$) and an ordinal cumulative model (Bürkner & Vuorre, 2019) with `logit` link function ($\phi$ are the latent cumulative thresholds).

We propose to reformulate the projection problem by solving the KL minimisation in the latent predictive space as:

$$\mathrm{KL}\left(p\left(\tilde{\eta} \mid \mathcal{D}\right) \parallel q_{\perp}\left(\tilde{\eta}\right)\right) = -\mathbb{E}_{\lambda_*}\left(\mathbb{E}_{\tilde{\eta}|\lambda_*}\left(\log \mathbb{E}_{\lambda_{\perp}}\left(p\left(\tilde{\eta} \mid X, \lambda_{\perp}\right)\right)\right)\right) + \mathrm{C}, \tag{6}$$

where $p(\tilde{\eta} \mid \cdot)$ presents the distribution of the latent predictions.

### 3.1 Non-exponential family case

The distribution $p(\eta \mid X, \lambda)$ is model-dependent and in non-exponential family case does not, in general, have a nice closed form. We propose to approximate $p(\eta \mid X, \lambda)$ with a Gaussian distribution, a choice motivated by several reasons: 1) the potential boundaries on $\mu$ are enforced only via the inverse link $g$, $\eta$ itself is unbounded so that its support matches the support of a Gaussian, 2) the Gaussian distribution belongs to the exponential family and thus the fast to solve projection Eqs. (3) and (4) can be used, 3) the model on $\eta$ is typically additive, such that commonly chosen Gaussian priors on the latent parameters $\lambda$ make the implied distribution of $\eta$ closer to Gaussian as well. Now that the transformed projection is again in the exponential family, we can apply the usual approach outlined in Section 2 to solve the projection. As the details of the complete projection predictive approach have been presented and evaluated elsewhere (Piironen & Vehtari, 2017a; Piironen et al., 2020b; Pavone et al., 2022; McLatchie et al., 2023), we focus here on evaluating the performance of the latent space projection.

Although the dispersion in the latent space is typically not a model parameter, we still need to approximate it to compute its projection in Eq. (4). The approximate dispersion is model dependent, and may sometimes even be known analytically. In ordinal cumulative family, the known latent dispersion is 1 by construction. If no analytical solution is available, approximation based on second derivatives can be used.

### 3.2 Exponential family case

The latent projection formulation also works for models in the exponential family. For Gaussian observation models, it coincides exactly with the original framework, and therefore no improvement is gained with the latent approach.

For non-Gaussian exponential family models, the original framework computes an approximate solution to Eq. (1) via PIRLS (as in Eq. (3)), which often results in unstable solutions for models with complex

structure or link functions (Catalina et al., 2022). In contrast, the latent approach computes an approximation on the latent space, removing the complexity of the link function and the response model. Experimental results show that the latent approach results in significant improvements also for non-Gaussian exponential family models.

## 4 Related work

For models in the exponential family, variable selection has a strong presence in the literature. Some methods perform variable selection by optimising a penalised likelihood formulation (Tibshirani, 1996; Zou & Hastie, 2005; Friedman et al., 2010a; Candes & Tao, 2007; Breiman, 1995; Fan & Li, 2001), while at the same time trying to select a subset of relevant variables. These methods impose a penalisation on large coefficients, effectively driving some of them towards zero, depending on the choice of regularization. For more details, we refer the reader to Hastie (2015). These methods suffer from several drawbacks. First, by dealing with the estimation of the model and the selection of variables at the same time, they often result in suboptimal solutions (Piironen & Vehtari, 2017a). Second, they are derived for specific likelihoods, and generalising them to other models is difficult, if at all possible. Most notably, these methods cannot perform variable selection on group-specific parameters in hierarchical models (Catalina et al., 2022).

These approaches have been generalized to specific likelihoods outside the exponential family, such as ordinal or Cox models (Wurm et al., 2017; Archer & Williams, 2012; Archer et al., 2014; Fan et al., 2005), but no general framework exists. Penalised likelihood approaches can be applied to models outside the exponential family by approximating the likelihood with an exponential family distribution. This enables variable selection, but the resulting model is likely to underperform in terms of predictive performance. A model in the original response space can be obtained by fitting a Bayesian model including only the selected variables. We compare our method against baselines that follow this approach in Section 5.2.

Variable selection is also addressed from the Bayesian perspective (O'Hara & Sillanpää, 2009). This is typically done by imposing so-called sparsity priors, such as the horseshoe (Carvalho et al., 2010; Piironen & Vehtari, 2017b) or the spike-and-slab (Ishwaran & Rao, 2005). As the posterior itself is not fully sparse, a sparse solution for variable selection can then be obtained by thresholding based on posterior expectations. Hahn & Carvalho (2015) discus related decopuling shrinkage and selection, that loosely follows the original decision theoretical idea by Lindley (1968). Piironen & Vehtari (2017a) and Piironen et al. (2020b) discuss the relationship of Hahn & Carvalho (2015) approach to projection predictive inference and discuss why the latter is more principled by following the original approach by Lindley (1968) more closely.

If a Markov chain Monte Carlo (MCMC) algorithm (Robert & Casella, 2013) is used for inference, no strong assumptions on the likelihood or overall structure are needed. This makes Bayesian inference applicable to models outside the exponential family with multilevel (or other complex) structure. This helps with the generality of the inference, but still falls short on the selection, since the user still needs to manually decide which variables should be selected.

Bayesian reference models have been used for variable and structure selection tasks in the context of exponential family models, including (additive) multilevel models and discrete response families with finite support (Piironen et al., 2020b; Catalina et al., 2022; Pavone et al., 2022; Piironen & Vehtari, 2016; Weber et al., 2023). However, its application to models outside the exponential family has remained unexplored.

Our approach solves variable selection on non-exponential family models by performing the selection on the latent space of the original model, therefore keeping the original structure. We approximate the unknown latent distribution with a Gaussian distribution. The problem of learning an implicit distribution has been approached in the Bayesian inference literature from different angles. Some authors tackle this problem with variational inference (Blei et al., 2017), by learning an implicit mapping between samples of the implicit distribution and a powerful and expressive learner, typically normalising flows (Rezende & Mohamed, 2016; Huszár, 2017; Pequignot et al., 2020; Titsias & Ruiz, 2019). Optimising these flexible models is typically expensive, requiring many iterations and diagnostics to have any guarantee of convergence (Dhaka et al., 2020). On top of that, the main bottleneck in projection predictive inference is not solving this projection

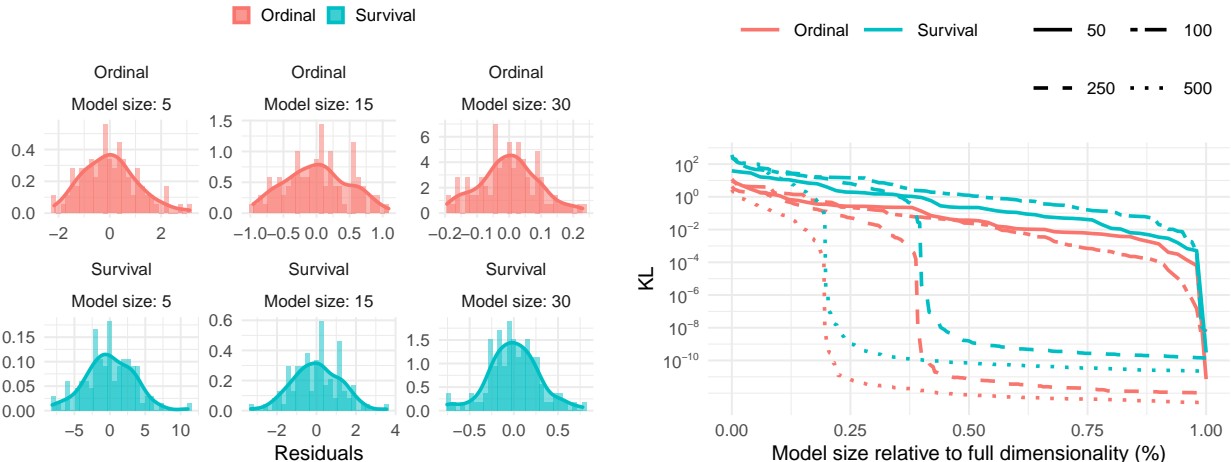

(a) Residual histogram plots for three selected projections for a reference model of size $D = 50$. In all cases, the residual distribution is close to normal with a decreasing standard deviation as more terms are introduced in the projection. Notice the shrinking $x$-axis as the projections include more terms.

(b) KL divergence between reference and projection predictive distributions. Different line types indicate size of the reference model. As projections include a higher proportion of the reference model, the KL-divergence approaches 0 faster.

Figure 2: Residual histograms and KL-divergence plots for projections on a simulated time-to-event survival analysis model and an ordinal cumulative model, both with $N = 100$ observations, and correlation factor $\rho = 0.3$.

*once*, but possibly many times, as complex models require many posterior draws to be projected for the projections to fully capture the posterior uncertainty in the reference model.

## 5  Experiments

We evaluate the performance of the proposed method with simulated and real-world data experiments. In particular, we must account for complex high dimensional posterior geometries, the role of correlated terms, and the size of the selected subset of variables in different models outside the exponential family.

First, we demonstrate the use of different diagnostics for the latent approximation. Second, we demonstrate the variable selection itself in high dimensional non-exponential family examples. Then, we extend the experiments to real datasets. Finally, we show the benefits that our method brings to models in the exponential family, too.

For the experiments, we used modified `projpred` (Piironen et al., 2020a) available at `https://github.com/stan-dev/projpred/tree/e1913e4b59cff00f1e7a5386c068431aa3368dec`. The latent space projection is now also available in `projpred` package since release 2.4.0.

### 5.1  Diagnosing the quality of the latent approximation

We use two diagnostics to assess the quality of the approximate projections:

- We perform projection predictive checks on the residuals $\tilde{\eta}_* - \tilde{\eta}_\perp$, where $\tilde{\eta}_*$ corresponds to the latent predictions of the reference model and $\tilde{\eta}_\perp$ to the predictions of the projection.
- We check the Kullback-Leibler divergence between the reference's and projection's predictions after convergence, which should approach 0 as more terms are included in the projection.

We use simulated data from 1) an ordinal cumulative model with `probit` link function and 2) a time-to-event survival model with a `log` link function. The outcome in both cases is generated as a function of

$D \in [50, 100, 250, 500]$ sampled predictors with a uniform correlation factor of 0.3, where only 60% of the predictors have a non-zero effect on the response. The full generation process for these data is detailed and further analysed in Section 5.2.

**Projection predictive checks assess normality assumptions.** Histograms of projection residuals (Fig. 2a) for various model sizes are a practical diagnostic for the normality assumption. As more terms enter the projection, the residuals get smaller and more concentrated, as indicated by the shrinking $x$-axis in the figure.

**Kullback-Leibler divergence shows that the latent projections eventually match the reference model predictions.** Even for the most challenging scenarios, the KL-divergence of the latent predictions shows that the projection predictive distribution gets closer (the KL-divergence approaches 0) to the reference predictive distribution as more terms enter the projection (Fig. 2b).

## 5.2 Non-exponential family models with simulated data

We compare the predictive performance of the optimal submodels in terms of held-out expected log predictive density (ELPD; Vehtari & Ojanen, 2012) for two types of models: an ordinal cumulative model and a time-to-event survival analysis model with a Weibull hazard process. Additionally, we examine the performance regarding the selection of truly relevant variables.

For the simulated high-dimensional data, we compare the performance of our approach to other popular sparsity promoting solutions in the literature:

- Elastic net regularization as implemented by `glmnet` (Friedman et al., 2010b), abbreviated as `glmnet`.
- Spike-and-slab priors as implemented by `spikeSlabGAM` (Scheipl, 2011), abbreviated as `ss`.
- Spike-and-slab LASSO priors as implemented by `SSLASSO` (Rockova & George, 2018), abbreviated as `sslasso`.
- Projection predictive inference on the approximate response space as implemented by `projpred` (Piironen et al., 2020a), abbreviated as `projpred`.

Since these approaches do not exist for the specific likelihoods we use, or for models outside the exponential family in general, we run variable selection on an approximate model, where we assume a normal likelihood of the response rather than the appropriate likelihood (cumulative or Weibull in our examples). Note that this normal approximation is different from our latent approach, since the latter approximates the latent predictor, not the response.

We first fit ordinal and time-to-event survival regression models on extensive simulation conditions. The generative process is

$$x_n \sim \text{Normal}(0, \Sigma_\rho), \quad z_d \sim \text{Bernoulli}(0.6),$$
$$\lambda_d \sim z_d \cdot \text{Normal}(0, 1.5), \quad \eta_n = x_n^T \lambda, \quad y_n \sim p(g(\eta_n), \phi),$$

where $y \in \mathbb{R}$, $x_n \in \mathbb{R}^D$ are the outcome and predictors respectively, $\lambda$ denotes the unknown coefficients, number of variables $D$ is varied in $[50, 100, 250, 500]$ and the number of observations $N$ is also varied in $[100, 200, 300]$. $\rho$ indicates the uniform correlation between predictors, $g$ is the model-specific inverse link function and $p$ is either an ordinal cumulative likelihood (with 5 ordered categories) or a time-to-event likelihood with Weibull base hazard process (maximum time-to-event is capped to 5), and $\phi$ are model specific parameters. Here, we show results for $N = 100$ observations and $\rho = 0$.

We fit the reference model making use of `brms` (Bürkner, 2018) with a regularized horseshoe prior (Piironen & Vehtari, 2017b) accounting for the sparsity in the predictors. The regularized horseshoe prior helps in avoiding overfitting, particularly for the models with a larger number of covariates.

Both the KL and residuals diagnostics indicate a good approximation for these models as shown in Section 5.1. Here we analyse the predictive and selection performance.

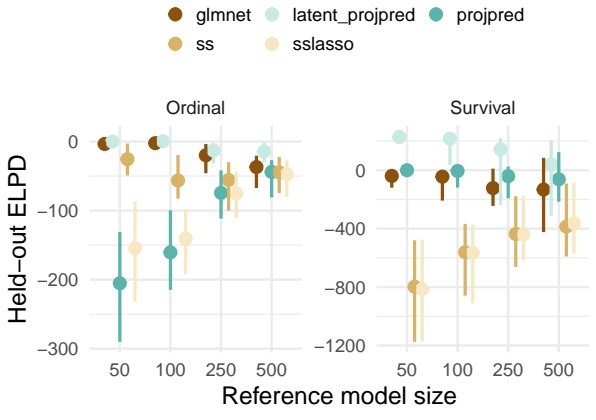 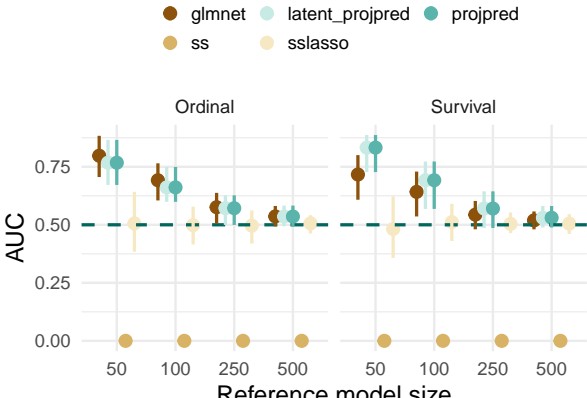

(a) Median and 90% confidence interval ELPD difference to the reference model for the selected subset of predictors for an ordinal cumulative model (left) and a time-to-event survival model (right) with $N = 100, \rho = 0$. Higher values are better. Competing methods are described in Section 5.2.

(b) Median and 90% confidence interval of the area under the curve (AUC) for the selection of truly relevant terms in an ordinal cumulative model (left) and a time-to-event survival model (right), with $N = 100$, and $\rho = 0$ for all competing methods. Higher values are better. Performance of pure random selection is shown as dashed dark green line. The methods are described in Section 5.2.

Figure 3: Mean and median values for ELPD difference and AUC for the selected submodels.

**Latent projections achieve the best held-out ELPD performance.** Fig. 3a shows that the latent approach always achieves the best performance, even surpassing the full reference model with models that include substantially fewer terms. This can be explained by a slight overfitting in the reference model. In the most complex scenarios, we see that the performance of all methods suffers from slightly larger variance, including the full reference model.

**Approximate likelihood methods result in underperforming selections.** While the performance of `glmnet` is the closest to our proposed method on average, its performance distribution is wider and less reliable, while other methods suffer greatly from the approximate likelihood representation (Fig. 3a). Spike-and-slab priors (`ss` in the figures) impose a very strong penalisation, which, together with the arbitrary choice of threshold (0.5 in our experiments, as is common, e.g. in Scheipl (2011)), can result in suboptimal selections. The ill-informed approximate likelihood results in very few terms crossing the selection threshold, particularly in `ss` and naive `projpred`. Even though `sslasso` does not need an arbitrary threshold for the selection, it suffers from similar problems.

**Other approaches require refitting a Bayesian model.** Other approaches operate on an approximate likelihood model that does not result in a sensible predictive model. To have a functional reduced model in the same domain as the reference model, all approaches except ours require refitting a full Bayesian model with the selected predictors and correct (non-normal) likelihood, increasing their cost. Our latent approach fully honours the reference model structure, and predictions in the original model space only require evaluating the inverse link function for the latent predictions.

**Latent approach retains reference model uncertainty.** The uncertainty present in the reference model is projected as well, contributing to better informed solutions than the other approaches, which cannot represent the original model likelihood. Fig. 3b shows that both `projpred` approaches achieve the highest AUC, even though naive `projpred` fails to assess the predictive performance of the projections. For the simpler `ordinal` experiments, `glmnet` achieves similar AUC. Interestingly, `ss` has trouble identifying any relevant features, and therefore its selection is not accurate, whereas `sslasso` manages to identify at least some.

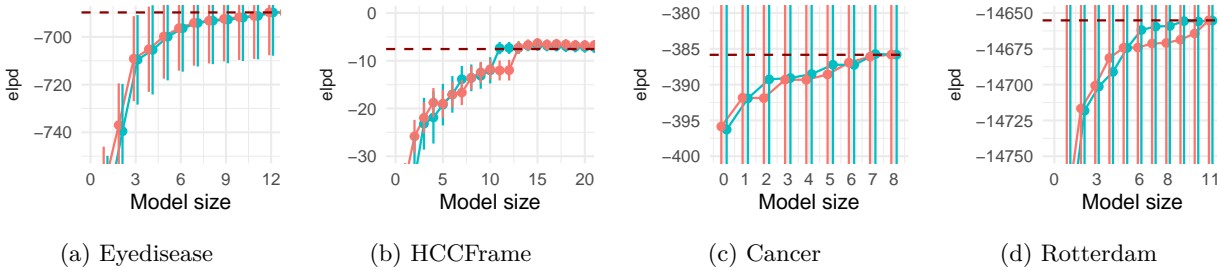

Figure 4: Full data zoomed-in ELPD performance comparison in the response model space. For `latent_projpred`, the projections are computed in the latent space, and the predictions are then transformed. Red shows `glmnet`, and cyan shows `latent_projpred`. Error bars indicate ELPD 95% quantiles. Larger error bars correspond to datasets with fewer observations.

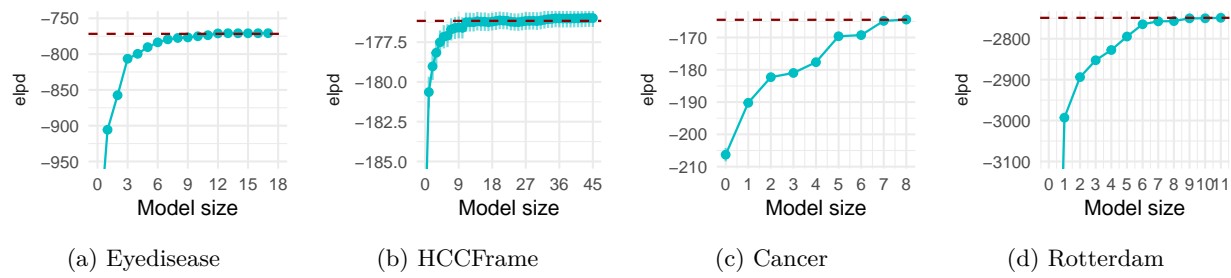

Figure 5: Full data ELPD performance in the latent predictor space for `latent_projpred`. The reference model performance is shown with dark red dashed horizontal line.

### 5.3 Non-exponential family models with real data

In this section, we assess the selection and predictive capabilities of our latent approach on real datasets and compare it against `glmnet` as the best competitor in our simulated examples. We focus on the quality of the selection path and the properties of the latent projections.

**Latent projections identify superior solution path.** Fig. 4 shows the comparison of the latent approach against `glmnet`. From a predictive performance point of view, the latent approach identifies a better solution path for all datasets we tested, except for `eyedisease`, where both approaches are equal. Note that, along the path, there are multiple model sizes for which the latent approach's projections' performance is superior to their `glmnet` counterpart. Datasets with few observations (`cancer` and `rotterdam` particularly) suffer from higher variance in performance.

**The response space is noisier.** Our latent approach allows us to perform the variable selection on either the response scale or the latent predictor scale. When compared to the selection in response scale ( Fig. 4), the latent predictor space is significantly less noisy (Fig. 5), mostly due to the complex observation model and sometimes relatively small datasets. The reference model filters noise from the data and thus the latent variables have less noise, resulting in reduced variance in the selection criterion (Fig. 5), even in data with few observations (`Cancer` and `Rotterdam` in particular).

### 5.4 Non-Gaussian exponential family models with simulated data

We simulate 50 realizations of a Bernoulli data with $N = 300$ observations and $D = 50, 100, 250, 500$ uncorrelated variables.

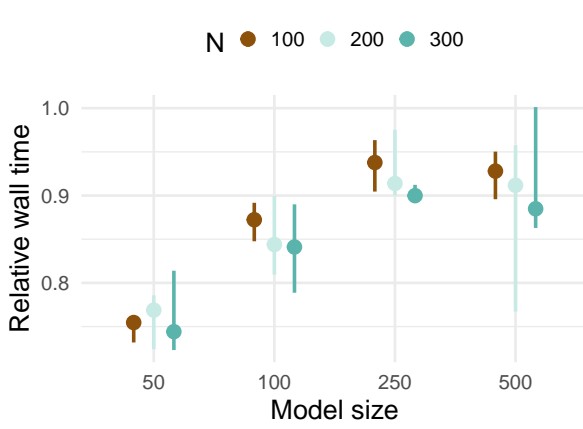

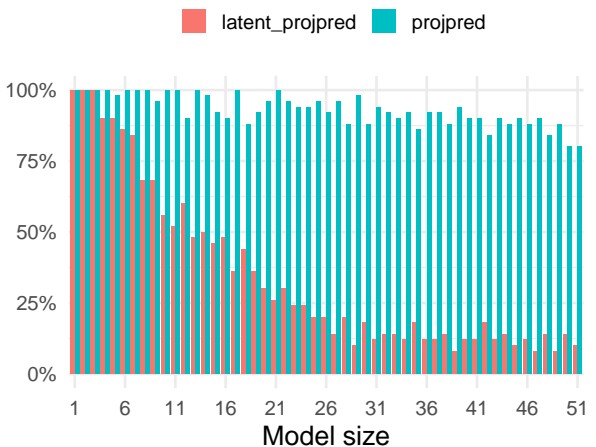

(a) Median and 90% interval of relative wall time of `latent_projpred` over `projpred` for 50 Bernoulli data realizations and varying number of observations (different colors). Lower than 1 means `latent_projpred` is faster.

(b) Bootstrap inclusion frequencies, for a simulated Bernoulli dataset with $N = 100$ observations and $D = 50$ variables, computed from 50 bootstrap samples.

Figure 6: Comparison of `latent_projpred` and `projpred` with simulated Bernoulli data.

**The latent approach is faster to compute.** The latent normal approximation is solved efficiently in closed form, which is faster to compute than the iterative solution of the original projection approach, resulting in significantly faster solutions (Fig. 6a).

**Latent approach has less variability and selects fewer variables.** We simulate 50 bootstrap samples from the same Bernoulli data generation process with $N = 100$ observations and $D = 50$ variables and compare the selection of the original framework against the latent approach. Fig. 6b shows that the latent approach results in smaller subsets of relevant variables with a smaller variability across bootstrap samples, while the original framework fails at identifying a suitable subset and overselects variables in every case. The difference in the behavior is mostly due to the model size selection rule. As seen in Figures 4 and 5, latent projection has smaller uncertainty on the submodel performance and thus there is less variation in the selected model sizes.

### 5.5 Non-Gaussian exponential family models with real data

For complex non-Gaussian models, the original framework provides a solution that might suffer from unstable projections and often fail to converge to sensible results. We show that the latent approach obtains a superior solution for a real model in the context of endangered species conservation taken from an ongoing collaboration with domain experts (Digby et al., 2023).

These data consist of a set of 211 kākāpō individuals (it is not possible to obtain more observations due to the very nature of the data set) and the aim of the study is to find a model for the fertility of upcoming eggs. We constructed a hierarchical Bernoulli model (the response is 0 for fertile and 1 for infertile eggs) with a varying intercept per individual. In short, the model can be summarized as

$$y_i \sim \text{Bernoulli}(\mu_i), \quad \mu_i = \text{inv\_logit}(\eta_i), \quad \eta_i = \alpha + X\lambda + z_i,$$

where $X$ is the design matrix containing all covariates, $\lambda$ are the population parameters, $\alpha$ denotes the population intercept and $z_i$ stands for the varying intercept for the $i$th individual. The group effect parameters can dominate the outcome variance in the model, which paired with the nonlinear link function, can often result in PIRLS solver convergence issues.

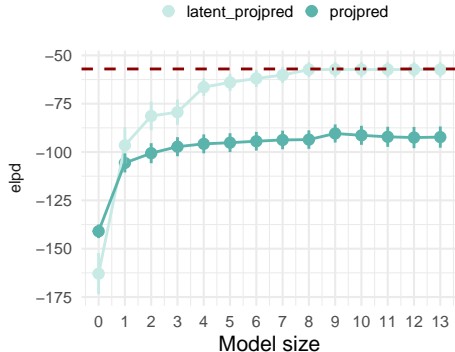

Figure 7: Comparison of full data expected log predictive density (ELPD) performance for a hierarchical Bernoulli model of fertility among kākāpō population. Reference performance in dark red dashed horizontal line. The original `projpred` solution does not reach the reference model predictive performance due to convergence issues in the underlying projections when including the varying intercept per individual.

**Latent approach provides superior performing projections.** Fig. 7 shows that the latent approach results in more stable and efficient projections for kākāpō fertility model. The latent approach accurately captures the reference model predictive performance, and also a better ordering in the solution path.

**A note on the convergence issues for original `projpred`.** As reported in lme4 issue tracker [12], it seems that the underlying solver for the projections in `projpred` may have issues in the case of non-integer responses for Binomial or Poisson likelihoods. Our investigation and diagnosing of the issue seems to indicate that the underlying cause would be in the solving of the integral for estimating group-effect parameters (Catalina et al., 2022; Bates et al., 2015). Models suffering from this issue show the estimated effects shrinking towards zero. This is a natural effect in binomial models, since responses closer to 0.5 indicate inability of discerning both groups. However, the issue present here is that the group-effects estimates are also shrunk towards zero already with response values are as high as 0.9 (quite close to 1), which in the end are translated into worse projections. These worse projections can partially explain some of the differences between `latent_projpred` and original `projpred`, and we must be cautious when drawing conclusions from these results only. It is also noteworthy that the latent approach did not show such issues.

## 6 Discussion

We proposed a novel latent space projection predictive approach to perform variable and structure selection in models with non-exponential family observation model. As shown in the experiments, the approach is also beneficial for non-Gaussian exponential family models.

We have shown that our approach offers superior performance in extensive experiments on high dimensional generalized linear models and several real datasets. The proposed method is not limited to latent linear models, as it can readily be used for variable and structure selection in more complicated models, including linear and non-linear hierarchical models (as demonstrated in Fig. 7).

Our approach not only improves variable and structure selection in a whole new set of models while respecting their original structure, but also offers superior performance in currently supported models in `projpred`.

The method may have worse performance for models whose latent space is far from normal. We provide diagnostics that alert the user in these cases, but further research is needed to improve the performance in such cases.

---

[1] https://github.com/lme4/lme4/issues/682
[2] https://github.com/lme4/lme4/issues/180

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
