# OpenReview forum: "Latent space Projection Predictive Inference"
_TMLR — Rejected by TMLR_

### Review · Reviewer_wAtd · 2023-12-04

**Summary Of Contributions:**

The authors consider projection predictive inference (PPI), but performing the approximate distribution matching in the latent predictor space rather than the response space. Promising results are presented for ordinal and time-to-event models using both simulated and real data, in comparison to the original PPI approach and typical variable selection approaches.

**Audience:**

Yes

**Claims And Evidence:**

Yes

**Requested Changes:**

Intro/background needs major rewriting/work.

Should compare to penalized regression using the same/correct likelihoods.

**Strengths And Weaknesses:**

I found the paper very hard to follow. The motivation is unclear: is the interest in prediction itself or parameter estimation? How does variable selection come about as part of this procedure without any explicit regularization? (some form of Occam's razor maybe?) q is a distribution, why is it a "projection"? The setup/background feels like it written assuming the reader is well-acquainted with PPI but it is, AFAIK, not a widely used approach. One very helpful step might be to provide an explicit running example (e.g. ordinal regression). Another would be to include a pseudocode/pseudoalgorithm/a receipe for analysis in this framework, e.g. 1) fit the reference model with MCMC and store posterior samples 2) estimate Lambdas for each posterior draw 3) form the predictive distribution based on these Lambdas.

My other main criticism is saying that methods don't exist to fit e.g. ordinal regression with L1 regularization, so just using a normal likelihood. First, I'm not convinced that is true. Second, even if it is it would be trivial to implement this with pytorch or TF and Adam.

In eq 1, expectations should be wrt to a distribution, not RVs. e.g. lambda* ~ p(lambda*|D) would be ideal. This expression is being minimized wrt q, specify that.

I think it is assumed q(tilde y)=mean(p(tilde y|lambda^(s)_orth)), specify that.

In eq 5 what are example of p(eta|lambda,X). Don't we usually just have e.g. eta=X lambda?

pg 4: "thresholding based on posterior"... if we just care about prediction why threshold?
decopuling

In 5.2 is beta=lambda? If so just use lambda. Also beta ~ z . Normal is v informal. This should be beta ~ pi Normal + (1-pi) delta_0 where pi = 0.6.

Define ELPD.

---

> ### Comment · Reviewer_wAtd · 2023-12-21
> **Writing**
>
> I read over the other reviews. We are all in agreement that the methods description was hard to follow to the point that we don't really know what they are doing. I hope the authors will be able to provide a clearer manuscript for us to digest.

---

### Review · Reviewer_G6HY · 2023-12-20

**Summary Of Contributions:**

In this paper, the authors propose a new method for projection predictive inference that increases its applicability to a large array of generative models. Specifically, the authors propose doing projection predictive inference in the latent predictor space. Empirical results demonstrate the applicability and superiority of the proposed method compared to baselines.

**Audience:**

Yes

**Claims And Evidence:**

Yes

**Requested Changes:**

More details are needed for understanding the proposed approach. *Critical*

**Strengths And Weaknesses:**

# Strengths

The idea is very clever and nifty. Moreover, the experiments section is *extremely* thorough.

# Weaknesses

I want to preface this by saying that I am not an expert in projective prediction inference.The biggest weakness is the writing. Specifically, while I understand the high-level idea, it is extremely hard for me to figure out what exactly the authors are doing. To start, what exactly is $q_{\perp}(\lambda)$? At first, I thought it was an approximate distribution but that doesn't seem to be the case. Is it the distribution of dimensions of $\lambda$ that have been selected from the procedure? Or is it the projection of $\lambda$ onto a lower-dimensional space using a matrix, $W$?

Next, in section 2 the authors state `As the integrals involved are in most cases computationally infeasible, Goutis & Robert (1998) suggested changing the order of integration and minimizations and solve the optimization for each posterior draw
separately`. What happens after the each optimization problem is solved separately? Are the resulting solutions averaged together? Moreover, can one interchange minimizations and expectations?

---

### Review · Reviewer_Gp67 · 2023-12-21

**Summary Of Contributions:**

The paper proposes a variable selection technique for Bayesian probabilistic prediction models. While related existing works have developed techniques for exponential family models, the paper develop techniques for some non-exponential family models, such as those in survival time regression or ordinal regression. Numerical experiments indicate that the proposed method finds variable subsets that lead to good models (measured in expected log likelihood on held-out data).

**Audience:**

Yes

**Claims And Evidence:**

No

**Requested Changes:**

# Primary

I recommend the authors rewrite Sections 2 and Sections 3. The explanation of projection predictive inference should be self-contained, rather than assuming that the reader is already aware of the technique's details. The setup for the proposed technique (class of probabilistic models considered) should be described in more detail, with concrete examples. The proposed technique itself should be described with some equations, even if the point is an eventual reduction to the standard exponential family case considered in existing works.

# Secondary

In plot b of Figure 2, clarify the x-axis scale. The ticks go from 0 to 1. But the label says (%). Should we interpret the 1 value as 1% of the reference model size, or 100% of the reference model size?

In plot a of Figure 2, you might constrain the different panels to have the same scale on the x-axis. It will make it easier to see that the scale on the x-axis reduces as model size increase.

Plots c) and d) of Figure 5 should be rescaled - the error bars span the entire y-axis, making it difficulty to read and assess which method is performing better.

# Time-permitting

Prediction quality is currently measured by expected log density. Are there other notions of quality relevant to the chosen data analyses? Maybe mean squared error for regression, or classification error for classification?

How does the proposed method compare with existing frequentist techniques, like glmnet, in terms of runtime?

Most experiments are conducted with dataset in the sizes of a few hundred observations. Would there be any conceptual issues with applying proposed technique to larger dataset?

**Strengths And Weaknesses:**

# Strength

Figures are well-labeled, have easy-to-read font sizes, and informative captions.

The paper has a good experimental setup to verify the quality of variable selection. By constructing synthetic data examples for which there is a notion of truly important features and testing the ability of different variable selection methods, the paper gives a convincing argument for the success/failure at the intended task.

The experiments in Section 5 cover a wide variety of situations. Section 5.2 and 5.3 in particular are good evidence for the claim that the proposed method can perform better than standard baselines.

# Weakness

The methodology is not clear from the writing in Section 2 and Section 3.

In particular, Section 2 reads like a summary of the methodology of related work, but does not have enough detail.
- Section 2 introduces much notation within the first few paragraphs, without concrete examples to instantiate the notation.
- In Equation (1), $p(\tilde{y} \mid \lambda_{\perp})$ is not defined anywhere else. The high level description "$q_{\perp}(\tilde{y})$ [...] be the projection predictive distribution" does not clarify for me what is the  meaning of $p(\tilde{y} \mid \lambda_{\perp})$.

Section 3 is the methodological key of the paper, but is not described in enough detail
- Equation (5) should have context and some examples. After reading the paper a few times, I have a sense that Equation (5) is meant to describe survival time regression / ordinal regression model. If so, the paper would strengthen if this point is made more explicitly.
- The equation right before Section 3.1 seems very important to the methodology, but also suffers from issues like Equation (1)
  - What exactly does $p(\tilde{\eta} \mid \lambda_{\perp})$ mean?
- The two paragraphs in Section 3.1 do not give a clear description of the a) the approximations involved and b) the computational procedure involved in the selection process.

---

### Author Response · Authors · 2024-02-01
**Response to reviewers**

Thanks for all the reviewers for the kind words and constructive suggestions.

General improvements:

- We have added clarifications to Sections 2 and 3.
- We have defined more notation, including $\lambda_\bot$, $q(\lambda_\bot)$, $q_\lambda(\tilde{y})$, $p(\tilde{y}|\lambda_\bot)$, $p(\tilde{\eta}|\lambda_\bot)$.- We have provided a summary of the complete projection predictive approach following the suggestion by Reviewer wAtd.
- Due to the space limit and existing review paper (McLatchie et al), we did not add full review of the approach. The focus of the paper is in evaluating the benefits of the latent space projection.
- We have made some of the figures bigger, but due to unfortunate family issues, we were not able to do make the improvements suggested by Gp67.

Below are some further responses to specific comments

> What happens after the each optimization problem is solved separately? Are the resulting solutions averaged together?

These are now clarified in Section 2

> Moreover, can one interchange minimizations and expectations?

Yes as an approximation, which we now mention explicitly.

> The motivation is unclear: is the interest in prediction itself or parameter estimation?

We have extended the first sentence of the second paragraph to: "We propose an efficient, stable, and information theoretically justified method to make variable selection for non-normal observation models in or beyond the exponential family, while retaining the predictive performance of the full model." and the following bullet points.

> My other main criticism is saying that methods don't exist to fit e.g. ordinal regression with L1 regularization, so just using a normal likelihood. First, I'm not convinced that is true. Second, even if it is it would be trivial to implement this with pytorch or TF and Adam.

To our best knowledge they don't exist, and we're happy to be learn if they exist and happy to get a pointer to how to implement them with pytorch, TF, Adam.

> In eq 1, expectations should be wrt to a distribution, not RVs. e.g. lambda* ~ p(lambda*|D) would be ideal. This expression is being minimized wrt q, specify that.

Both notations are commonly used, and we opt here to use the more compact notation.

> In eq 5 what are example of p(eta|lambda,X). Don't we usually just have e.g. eta=X lambda?

Due to the space limitations we are not discussing other options, like, Gaussian processes (Piironen and Vehtari, 2016).

> pg 4: "thresholding based on posterior"... if we just care about prediction why threshold? decopuling

Thresholding is something that others do for variable selection. We do not like it, but mention it when refering to related methods.

> Define ELPD.

The ELPD acronym is defined when used first time (in the beginning of
Section 5.2).

---

### Decision · Action_Editor_nymV · 2024-02-19

**Recommendation:** Reject

**Comment:**

The paper represents an honest effort, with clear motivations and discussion of the existing literature, plus initial experimental evidence suggestive of the utility of their approach.

On the other hand, even after an extended period for revision, all of the reviewers ended up recommending against acceptance of the paper in its current form. All of the reviewers initially raised serious issues of clarity regarding the exposition of the proposed method, and while the authors did make efforts to improve the manuscript, the opinion of all reviewers was not moved sufficiently in the direction of acceptance. Below, I will share an except from the three reviewers' final recommendation statements made to me.

1
> *The paper's methodological sections are not clear enough for the audience to understand the steps involved in the workflow.*

2
> *I thank the authors for updating the manuscript. While things are slightly clearer, I'm still having trouble following what is going on. While it I understand that this is based on the projection predictive approach I still think some sort of review is needed for the reader to understand what is going on. While there is a space restriction, adding more details or even an algorithm table to the appendix would have helped substantially.*

3
> *The exposition is a little clearer now but my other concerns haven't been addressed. L1 regularized ordinal regression methods are available, e.g. glmnetcr and ordinalNet in R, and easily implemented using https://github.com/EthanRosenthal/spacecutter in python...*

Having looked through the revised paper myself, it is clear that the authors have made an effort to improve their exposition, but I am in agreement with the reviewers that a rather substantial amount of work is still required to ensure that the proposal here is presented in a clear enough manner to be digested by the wide audience that it has in mind. As such, my decision is rejection, with the option open to the authors to submit a major revision at a later date.

**Audience:**

The problem setting considered by the authors is general, and the desire to develop efficient/effective variable selection methods under a wider variety of underlying model distributions is perfectly natural. As such, in principle the paper should have a wide audience in the TMLR community.

**Claims And Evidence:**

In this work, the authors are interested in variable selection for learning problems within a Bayesian framework. In particular, the direct context in the literature for this work is past research on "projection predictive inference," in which only a subset of the rich "reference" model parameters are used. Which parameters are to be kept is typically determined by minimizing the KL divergence between distributions before/after culling parameters. In the case where the distributions being measured are members of the exponential family (depends on both functional form of predictors and distributional assumptions), then a practically congenial maximum likelihood type objective function can be readily derived, though this does not extend (in general) beyond the exponential family. To generalize this methodology, the authors re-formulate the objective in terms of a latent predictor, and take the approach of approximating the distribution of the latent variables (conditioned on inputs and parameters) by a Gaussian distribution, which puts the problem back in the domain of exponential models, meaning existing insights for projection predictive inference from past work can be applied. The authors discuss if/when assumptions of Gaussianity can be expected to be valid, and present numerical evidence that their approach leads to effective variable selection (as measured by expected log likelihood).

**Resubmission Of Major Revision:**

The authors may consider submitting a major revision at a later time.